# Aquaculture of Animal Species: Their Eukaryotic Parasites and the Control of Parasitic Infections

**DOI:** 10.3390/biology13010041

**Published:** 2024-01-11

**Authors:** Henry Madsen, Jay Richard Stauffer

**Affiliations:** 1Department of Veterinary and Animal Sciences, Faculty of Health and Medical Sciences, University of Copenhagen, Dyrlaegevej 100, 1870 Frederiksberg C, Denmark; 2Department of Ecosystem Science and Management, The Pennsylvania State University, University Park, PA 16802, USA; vc5@psu.edu; 3South African Institute for Aquatic Biodiversity, Makhanda 6140, South Africa

**Keywords:** aquaculture, parasites, diseases, zoonosis, invasive species, One Health

## Abstract

**Simple Summary:**

Aquaculture is very important for producing high quality food for the increasing human population, and humans’ dependency on it will increase further in coming years. Aquaculture, however, is also undertaken for several other purposes. Further development of aquaculture can be challenged by parasites, which can cause serious mortality among cultured species and incur costs of prevention and treatment of such infections and may render the final product unsellable. Of further risk are farmed species, which may act as a reservoir of parasites that threaten humans. Here, we review the eukaryotic parasites affecting various species used in aquaculture. These parasites are very diverse and of different significance in terms of morbidity.

**Abstract:**

Parasites are very diverse and common in both natural populations and in stocks kept in aquacultural facilities. For most cultured species, there are important bacteria and viruses causing diseases, but eukaryotic parasites are also very important. We review the various combinations of aquacultured species and eukaryotic parasitic groups and discuss other problems associated with aquaculture such as eutrophication, zoonotic species, and invasive species, and we conclude that further development of aquaculture in a sustainable manner must include a holistic approach (One Health) where many factors (e.g., human health, food safety, animal health and welfare, environmental and biodiversity protection and marketability mechanisms, etc.) are considered.

## 1. Introduction

Aquaculture is and will be important in feeding an increasing human population, but it is also undertaken to produce food for other species that are farmed. Aquaculture also includes the production of specimens for sport fishing; scientific research; the conservation and reintroduction of endangered, rare and extirpated species; bait used by anglers; ornamental trade [1]; and biological control of vector species. Aquaculture for biological control includes the production of animals that can be used to control nuisance species, such as certain species of aquatic macrophytes or vectors or intermediate hosts of zoonotic parasites such as malaria or schistosomiasis. Examples are *Gambusia* spp. or other fishes for the control of mosquito larvae [2,3]; *Macrobrachium* spp. [4,5,6], or fishes [7] for the control of schistosome host snails; grass carp, *Ctenopharyngodon idella*, for controlling aquatic vegetation [8]; and black carp, *Mylopharyngodon piceus*, for snail control [9]. The increasing demand for animal protein would lead to the overharvesting of natural populations and that could have serious consequences, as has been seen in Lake Malaŵi, where overfishing likely resulted in the transmission of schistosomes along open shorelines, which previously were considered free from transmission in the lake [10,11]. In that situation, aquacultural development has been suggested as a means of protecting near-shore molluscivorous fishes [12]. The most common species groups produced in aquaculture are shown in Table 1.

Various diseases and parasites can be a major limitation to the development and sustainability of aquaculture. Although parasites are an important part of biodiversity [13] and can help maintain stability in natural populations, they can also lead to extinctions [14]. Conditions in aquacultural facilities may be conducive for outbreaks of diseases caused by parasites, but can also make it difficult for some parasitic groups to establish [15]. Diseases may emerge through the exchange of pathogens with wild populations, evolution from nonpathogenic micro-organisms, and deliberate transfer of stocks [16]. Aquacultural practices often result in high population densities and other stresses such as poor water quality, which may increase the risk of the establishment and spread of infection [13]. As aquaculture expands and new species are included in aquaculture, new diseases may emerge and affect both wild and farmed fishes adversely [16]. 

**Table 1 biology-13-00041-t001:** Summary of most important taxa (+) produced in aquaculture in either inland or coastal-marine systems, with some statistics on the total production (in tonnes live weight in 2020 according to the FAO) [17].

Cultured Species	Inland (Freshwater)	Coastal-Marine	Inland	Coastal-Marine
Pisces	+	+	49,120,461	8,340,633
Mollusca			192,671	17,547,855
Gastropoda	+	+		
Bivalvia	+	+		
Cephalopoda		+		
Crustacea	+	+	4,477,201	6,759,815
Other			593,707	468,584
Corals		+		
Annelids	+	+		
Anurans	+			
Turtles	+			
Crocodiles	+			
Other	+	+		
Algae	+	+	593,707	35,013,089

Pathogens such as viruses and bacteria are very important in aquaculture [18,19,20,21], but eukaryotic parasites can also threaten aquacultural success and threaten native species if they are introduced or escape [22]. Some of the parasites associated with aquaculture may be zoonotic and hence a threat to human health, but could at the same time affect somatic growth and mortality of the cultured species [23]. Also, there are many pathogens or parasites that are non-zoonotic or only occasionally zoonotic and some of these could have important impacts in aquaculture, such as by causing high mortality, reduced somatic growth, and/or reproduction of the cultured species [23]. 

Parasitic infections in aquacultured animals have huge economic implications in reducing productivity and/or rendering products unsellable [24]. An important aspect is that poor culture conditions may cause stress and render the cultured species more susceptible to parasitic infections and disease [25]. Multitrophic aquaculture is a popular approach, but this could also promote transmission of parasites depending on the combination of species [26].

We review here the various combinations of aquacultured species and eukaryotic parasitic groups and discuss other problems associated with aquaculture such as eutrophication, zoonotic species, and invasive species, and we conclude that further development of aquaculture in a sustainable manner must include a holistic approach (One Health), where many factors (e.g., human health, food safety, animal health and welfare, environmental and biodiversity protection and marketability mechanisms, etc.) are considered.

## 2. The Major Parasitic Group

Depending on the species, type of culture (e.g., ponds, closed systems, cages, mono culture—multitrophic system) in either freshwater or brackish/marine water, and feed type (manure to commercially produced feed), many taxa may infect (ecto- or endoparasites) or be commensals, but not all species are important unless there are heavy infestations. These include various protistan taxa such as unicellular organisms and water molds, algae fungi, nematodes, platyhelminths (Monogenea, Digenea, and Aspidogastrea), and some species of mollusks (e.g., some marine snails and larval stages of some bivalves). An overview of the parasitic taxa in various cultured species is presented in Table 2, where we also attempt to rate each taxon’s relative importance. Obviously, such a rating is problematic and could be biased with higher ratings to parasites affecting species accounting for the majority of the aquaculture production, while parasites affecting the less commonly cultured species could be equally important at the farm level. Some parasites, e.g., some oligochaetes and leeches may be very important in natural freshwater populations but less so in aquaculture [15]. In marine aquaculture, parasitic levels may differ between shore and off-shore facilities [27].

The use of non-native species in aquaculture can be problematic as these species may carry new parasites or they could become invasive, especially if they have been genetically modified for fast growth and reproduction. Alternatively, native parasites could also adapt to use the introduced species as a new host. As new species are being selected for aquaculture, new parasite–host relationships may establish and become problematic [16]. Many parasites exist in natural populations and culture conditions may prove to be conducive for transmission.

### 2.1. Protista

The kingdom Protista is diverse and includes all eukaryotic organisms that are neither animals, nor plants, nor fungi [28]. Protists can be distinguished from plant, animal, and fungal cells by their ability to move on their own [28]. Most species are free-living, but some are parasitic [29]. The taxonomy of Protista has undergone considerable revision and terms used previously are no longer recognized as formal taxonomic groups, such as algae, zoosporic fungi, and protozoa [30]. A useful overview of the classification can be found in Ruggiero [31]. The kingdom is paraphyletic and many species parasitize animals or plants [32]. The species parasitizing aquatic animals include some protozoa, flagellates, amoebae, and ciliates belonging to these major clades: Apicomplexa, Ascetosporea (Haplosporidia and Paramyxida), Cercozoa, Perkinsozoa, Amoebozoa, Ciliophora, and Dinoflagellata [32,33,34,35,36]. Although some protistans utilize vectors and/or intermediate hosts in transmission, most transmit through contact between and within hosts [32]. The majority of protistans are, with a few exceptions, not a zoonotic concern, while they are very important for aquatic animals [32].

### 2.2. Myxozoa

Myxozoans are specialized cnidarian parasites [32]. Many species alternate between a vertebrate and an invertebrate host, but apparently some may not need a second host [32]. Infectious spores produced within a host typically proceed via cell-within-cell division, i.e., pluricellularity [32]. Invertebrate hosts, where sexual reproduction takes place, are typically oligochaetes, but also include Monogenea, polychaetes and bryozoans [32]. Mature actinospores are released from the invertebrate hosts and they can, upon contact, infect the vertebrate host. Vertebrate hosts in freshwater include fishes and amphibians, and for marine species, these are primarily fishes. The majority of myxozoans have been described from both freshwater and marine fishes [32].

### 2.3. Fungi 

Fungi contains some species that are important pathogens or parasites in aquaculture, such as the Oomycota or slime molds [32], which now may be considered as belonging to the Protista. Microsporidia were previously classified as Protista, but they are now considered to be basal fungi [37]. Fungi and fungal-like organisms have direct life cycles and some species may have resting stages outside their host [32]. Some fungal species may show some host specificity or tissue specificity [32]. Fungal and related taxa have been reported in all major animal groups as well as in plants and humans. The species infecting aquacultured species, however, are not normally harmful to human health [32].

### 2.4. Platyhelminthes

#### 2.4.1. Monogenea

Monogeneans are parasitic flatworms commonly found on the skin and gills/buccal cavity of fish and within the urogenital system of anurans and chelonians [32]. They can also to a lesser extent be found in other locations in fishes, such as the nares, stomach, coelom, heart, and circulatory system [32]. 

Monogeneans are typically 0.3–20 mm long but can vary from about 200 µm to 4 cm, and they have a high degree of host specificity [32]. By means of a specialized muscular posterior attachment organ called an opisthaptor, they attach to their host [32]. The opisthaptor has a variable array of sclerotized hooks or other structures, and the number and shapes of these structures are characters used in identification of species together with characters of the copulatory organs [32]. Most monogenean species have a single definitive host in their life cycles, but immature stages of a few species may have an intermediate host. Monogenean species do not pose a human health risk.

#### 2.4.2. Digenea

Digeneans, often referred to as flukes or digenetic trematodes, are endoparasitic flatworms, principally infecting the alimentary canal or associated organs [32]. Digenea belong to the class Trematoda, which also includes the Aspidogastrea (which use a mollusk and possibly a vertebrate host). Several digenean species are of great public health importance by causing serious morbidity in humans; these include species of *Schistosoma* [10,38,39], and several fish- or other food-borne trematodes [40,41].

Depending on species, digenetic trematodes may need 2–4 hosts to complete their life cycle [32]. Most vertebrate species can serve as final hosts for at least some trematodes and most species use a mollusk, primarily gastropods, but bivalves can also be intermediate hosts for some species. The first intermediate hosts become infected either by ingesting eggs released from the final host or via the miracidia hatching from eggs actively penetrating them. Cercariae released from the first intermediate host can either infect the final host directly (e.g., schistosomes) or infect a second intermediate host, in which they form a resting stage, metacercariae, awaiting for the host to be consumed by the final host (e.g., food-borne trematodes) to continue their development. Many species, including mollusks, crustaceans, and fishes, may serve as second intermediate host for many trematode species while for fasciolid species, metacercariae lodge on plants. If fish are the final hosts, the adult digeneans are, with some exceptions, localized in the gastroenteric apparatus [32]. If fish are second intermediate hosts, the metacercariae lodge in several organs, including the eyes, gills, skin, musculature, heart, and kidney.

#### 2.4.3. Cestoda

Tapeworms are endoparasites (primarily intestinal) of vertebrates and they display a high degree of host specificity, with each species infecting a single definitive host species or group of closely related host species. They have a worldwide distribution. Besides the vertebrate definitive host, cestodes require at least one or, more frequently, two intermediate hosts [32]. These intermediate hosts are often arthropods but can also be vertebrates depending on the species.

Adult cestodes have a flattened body with a scolex (grasping head), which attaches the cestode to its host’s intestinal wall, a short neck, and segmented trunk formed of a variable number of hermaphroditic proglottids (strobilum) [32]. Cestodes absorb nutrients from the host’s alimentary tract and produce a large numbers of eggs. Fishes can serve either as an intermediate host or as a definitive host [32]. The life cycles of cestodes are complex and variable among species and therefore infection opportunities in aquacultural environments are rather limited, with just a few species being of concern [32]. This is in contrast to wild fish, which often are infected with cestodes and therefore could pose a risk to farmed aquatic hosts [32]. Many species of cestodes have public health implications [32].

### 2.5. Nematoda

Nematodes are diverse and inhabit a broad range of environments. Most species are free-living, but many are parasitic. The free living species feed on micro-organisms and detritus and some species play an important role in the recycling of nutrients. Parasitic nematodes are a widespread group and include pathogens in most plants and animals. Nematodes can infect both wild and, to a lesser extent, farmed fish [32]. 

Most nematode species are dioecious but some species are hermaphrodites. Generally, the embryo develops to a first-stage juvenile and after hatching, it undergoes four molts before developing into the mature adult. In particular, mollusks and crustaceans can serve as intermediate hosts for the larval stages. In fishes, the adult nematodes are usually found in the intestinal lumen of their host. Fishes can serve as either an intermediate or definitive host [32]. In some cases, e.g., anisakid nematodes, many species of fish can also be a paratenic host [32]. Several nematode species, such as ascarids, filarias, hookworms, pinworms, and whipworms, parasitize humans and are therefore a public health concern. Some of these could be of relevance for aquaculture.

### 2.6. Acanthocephala

The acanthocephalans are a group of parasitic worms that are characterized by an eversible proboscis that is armed with spines, which is used to pierce and attach to the gut wall of its host [42]. The life cycle of acanthocephalans is complex and involves at least two hosts. Adult parasites are found within the intestine of their final host, where they reproduce sexually. Potential hosts may include invertebrates, fishes, amphibians, birds, and mammals. Acanthocephalans are sexually dimorphic [43]. Fertilized eggs develop inside the female uterus to the so-called acanthor stage and are then expelled and enter the aquatic environment with the host’s feces [32]. When ingested by a suitable intermediate host, usually a crustacean or (rarely) a mollusk, the eggs hatch into an acanthella stage, which encysts in the host’s tissues as a fully developed larval cystacanth until consumed by the final host [32]. Acanthocephalans have been occasionally reported in humans [42]. 

### 2.7. Gastropods

The Pyramidellidae has a worldwide distribution and is a large family of marine, ectoparasitic mollusks. The shell form varies from turriculate to ovate, with a variety of sculpture types, and usually they have a heterostrophic protoconch [44]. Pyramidellids pierce the skin of the host using their extensible proboscis, at the end of which is a stylet. They suck body fluids from their hosts, which include a wide variety of invertebrates, particularly polychaete worms [45]. Hosts may suffer various morbidities in response to infestation such as those reported from some economically important bivalves, i.e., shell malformation, lower growth rates, increased mortality, and increased transmission of bacterial disease [46]. Also, in general, Wentletraps (Gastropoda: Epitoniidae) are permanent ectoparasites or foraging predators that are mostly associated with corals [47,48].

### 2.8. Bivalve

Some species of freshwater mussels, especially in the order Unionida, parasitize fish as a means of dispersal [49,50]. The parasitic larval stage glochidia attaches itself to the gills or fins of the host fish, where it becomes encysted in the tissue, eventually after several weeks or months excysting to develop into a free-living adult [49]. Apparently, infestations of glochidia cause only minor effects on their hosts with the exception of heavy infestations, and the effects are typically only observable in combination with other stressors [49]. Unionids are severely endangered due to habitat destruction, introduction of non-native species, and loss of host fishes [51].

Glochidia of invasive species may have more serious effects on native fishes, e.g., as reported for *Sinanodonta* (*Anodonta*) *woodiana*, which cause a reduction in the condition factor of parasitized native species [52]. Experimentally, the infection of European Chub (*Squalius cephalus*) resulted in a reduction in the body mass and condition factor, and this was associated with changes in several physiological parameters measured in the plasma of host fish [52].

### 2.9. Arthropoda

Arthropods are a highly diverse, species-rich phylum. The Crustacea (subphylum) is very speciose (>64,000 species), many (>8000) of which have parasitic stages. These are primarily species of the taxa Copepoda, Branchiura, and Isopoda [32].

#### 2.9.1. Copepoda

Copepods are small crustaceans found in nearly all freshwater and saltwater habitats. Free-living copepods are important to global biodiversity, as the dominant part of zooplankton and an important food for other species, especially juvenile fishes. Many species have parasitic stages. Parasitic copepods generally have a direct life cycle [32].

Parasitic stages may be found moving over the external surfaces of their host, attached to the host by special attachment structures, or partially or completely embedded in the host tissue [32]. Copepods are dioecious and females extrude their eggs into paired egg sacs or uniseriate egg strings until they hatch [32]. Parasitic copepods have two life phases, i.e., a free-living, lecithotrophic naupliar phase and a copepodid phase, in which body morphology changes markedly and the animals switch to the parasitic mode of life [32]. 

#### 2.9.2. Isopoda

Isopods include both terrestrial and aquatic species; various feeding modes exist among these species, and some are internal or external parasites, mostly of fish. Adult isopods are flattened dorsoventrally and lack a carapace, and the head is fused to the first somite; they have sessile eyes and a body with seven somites, each with a pair of uniramous swimming legs [32]. Some species are obligate parasites (e.g., *Cymothoa*) and these can pose a serious problem in aquaculture, while other species are free-living scavengers, which occasionally can parasitize fish [32].

#### 2.9.3. Branchiurans

Branchiurans are obligate ectoparasites also known as fish lice. They are found primarily on marine and freshwater fishes but can also be found on invertebrates, salamanders, tadpoles, and alligators (Table 2). Branchiurans have a flattened oval body that is almost completely covered by a broad carapace. Adult branchiurans such as *Argulus* spp. are dioecious and very mobile, and they can leave their hosts temporarily and move between hosts to mate and lay eggs [32]. Some species feed on the blood of their host, while others feed on mucus and extracellular material. Their first maxillae are modified as large suctorial organs, and they have compound eyes, a preoral stylet, four pairs of swimming legs, and a tubular mouth [32]. 

#### 2.9.4. Pentastomida

Pentastomes are parasitic arthropods also known as tongue worms and they have a worldwide distribution [53]. The adults, which are large, elongated 0.5–12 cm, often tongue-shaped, and live in the upper respiratory tract of reptiles, birds, and mammals, where they lay eggs [53]. The eggs leave the host, either coughed out or through the digestive system, and these can then be ingested by an intermediate host, fish, or small mammal. After being ingested by the fish, which is the intermediate host, the egg hatches in the intestine, bores through the intestinal wall, and encysts in a suitable organ [53]. The cuticle of adults is transversely striated and the anterior end is thick and flattened with two pairs of strong hooked claws [53].

### 2.10. Fishes

A few taxa within the group can be classified as parasitic, e.g., some lampreys, which are jawless fish. The adult lamprey has a toothed, funnel-like sucking mouth [54]. They live mostly in coastal and fresh waters and are found in most temperate regions. Some species migrate great distances in the open ocean [55] while others are found in land-locked lakes. The larvae have a low tolerance for high water temperatures, and this may explain their absence from tropical waters [55].

The Great Lakes of North America support a large and profitable freshwater fishery, but this is threatened by parasitism from the invasive sea lamprey, *Petromyzon marinus* [56]. Invasion by the parasitic sea lamprey has contributed to major declines in many Great Lakes fishes’ populations [57]. Although the sea lamprey is an unwanted invasive species in the Laurentian Great Lakes, most lamprey species are of ecological and cultural value, and for some species, a conservation concern [58]. Few studies on parasites of lamprey have been conducted [59].

Although probably not important in aquaculture, another parasitic fish is the Candiru *Vandellia cirrhosa*, which is a small catfish from the Amazon which swims up the urethra of humans and parasitizes the human [60,61].

## 3. Main Animal Species Raised in Aquaculture

Aquaculture is very diverse both in terms of purpose and the species cultured. Species of many taxa are used in aquaculture and the production of all may be challenged by infections with various pathogens or parasites. As new species are selected for aquaculture, new parasitic problems may arise. The species causing problems depend on the cultured species and the severity of these organisms depends on the aquacultured species and the density of infection. Examples of production systems are shown in Figure 1.

Important species produced in aquaculture are summarized in Table 1 and these will be discussed below. Many aquacultural activities will show variation among regions (Table 3). According to the FAO [17], the total aquaculture production in 2020 comprised 87.5 million tonnes of aquatic animals mostly for use as human food, 35.1 million tonnes of algae for both food and non-food uses, and 700 tonnes of shells and pearls for ornamental use, reaching a total of 122.6 million tonnes in live weight [17]. The production of finfish is quantitatively the most important production, especially in inland culture [17]. The culture of mollusks and crustaceans, primarily in coastal and marine aquaculture, is also very important, especially in Asia [17]. 

For example, aquaculture of freshwater snails for food for people is especially important in South East Asia, but this is not limited to that area [17]. Aquaculture is practiced under very different conditions across the world, and many aquatic species and their hybrids are raised in different types of aquacultural farming systems using freshwater, brackish water, seawater, or inland saline water [17]. 

### 3.1. Corals

Corals are cultured primarily for ornamental aquaria or for the restoration of natural coral reefs [62,63]. Many coral reefs are under threat of degradation and this has led to increased research on active intervention strategies in an attempt to restore reefs and build ecosystem resilience [64]. Several studies have attempted to demonstrate the feasibility of large-scale coral aquaculture to provide a source of corals to transplant onto degraded coral reefs [64]. The parasitic species affecting corals were reviewed in Barton [64], and include some species of Acoela, Digenea, Polycladida, Gastropoda, Crustacea, and starfish. Some of the gastropods and crustacea may be more like predators than parasites [64].

### 3.2. Annelids

Several species of large annelids, such as the polychaetes *Arenicola* spp. and *Hediste* spp. are produced in aquaculture for fish bait or as feed for fishes or crustaceans in culture [65,66,67,68]. Collection of these and other annelids from the wild for bait has quite detrimental effects [69]. Annelids feed on bacteria and may be found at a high density at the sea bottom below cages used for culture of fish, and they are believed to be able to assist in the remediation of wastes from aquacultural facilities [70,71,72]. 

Annelids serve as intermediate hosts of myxozoans, which can be transferred to fishes and other groups that eat the infected annelid. Several species of annelids may also be parasitic, for example some species of leeches, *Hirudinea* [73,74].

Freshwater annelids such as *Chaetogaster* are common commensals of gastropods and bivalves both in natural and cultural conditions, but their effect may be minimal unless the infestation intensity is very high [75,76].

### 3.3. Mollusks

Mollusks, especially bivalves, are important, but gastropods, and cephalopods are also widely cultured for food for people [17], for feed for other species in aquaculture, for shells as collectors’ items [77], and for restocking natural populations [78,79].

#### 3.3.1. Gastropods

Gastropods are cultured for food in both freshwater and brackish/marine systems, while marine species may also be produced for the shells as collectors’ objects. Freshwater gastropods cultured for food (for people or as animal feed) mainly belong to Viviparidae and Ampullariidae because of their size. In South East Asia, these species are easily cultured either alone in relatively small tanks or ponds or in multitrophic fish ponds [26].

Parasites of snails include a wide range of species (Table 2) including Protista, fungi such as microsporidia, annelids (e.g., Hirudinea and *Chaetogaster* spp.), trematodes, nematodes, small crustaceans (Harpacticidae), Ostracoda, and others [75,80]. In particular, trematodes are important parasites of snails in both natural habitats and in aquacultural ponds. Trematodes can cause massive mortality among snails in natural habitats and especially damaging are trematodes with rediae stages [81,82]. Snails can also serve as second intermediate hosts for some food-borne trematode species and hence could play a role in the transmission of these trematodes if consumed insufficiently cooked.

Due to their role as first intermediate hosts for specific trematodes, some of which have major medical or veterinary importance, these species have received much attention. Many of these trematodes, including the zoonotic species, cause growth retardation, reduced reproduction, and increased mortality in the aquaculture.

#### 3.3.2. Bivalves

Both the marine and freshwater aquaculture of bivalves is important for food and pearls. Globally, the culture of bivalves is huge (Table 1).

Only a few fungi are pathogenic to bivalves and these mainly affect the external parts, such as the shell or byssus. The larval stages are especially prone to disease. There is one disease, however, shell disease, which has severe effects on *Ostrea edulis*, *C. gigas*, *Saccostrea cucullata*, and *Crassostrea angulata* [83,84].

Protistans are the most important cause of disease in bivalves. Most species belong to the phyla Haplosporidia, Dinoflagellata, or Cercozoa (*Marteilia* spp.) and infect mainly oyster and clam species. The main helminth parasites of bivalves are trematodes, cestodes, and nematodes (Table 2). Helminths cause relatively little disease in bivalves compared to protozoan parasites. A substantial amount of research is available in the literature on infections of helminths, partly because of their size and because they have complex life cycles with several hosts, and because in some cases they may infect humans through consumption of infected shellfish. Digenetic trematodes may use bivalves as the first intermediate host (sporocyst, redial and cercarial stages) or as the secondary host (metacercarial stage), or the bivalve may be the host for all stages of the life cycle. Trematodes that use bivalves as intermediate hosts may cause problems in aquacultured fishes [85].

Although nematodes are relatively uncommon parasites of marine bivalves, a few species do cause significant losses [86]. Several species of the polychaete annelid *Polydora* spp. burrow into the shells of bivalves and excavate U-shaped tunnels that subsequently become filled with compacted mud, causing ‘mudblisters’ on the shell. Some leeches may lodge in bivalves [87].

Compared with bacterial or protozoan parasites, crustaceans are only mildly pathogenic to bivalves. The best-documented disease organism is the copepod *Mytilicola intestinalis*, found in European waters [88]. 

#### 3.3.3. Cephalopods

Cephalopods are the most active and specialized class of mollusks. They may have a chambered shell (e.g., *Nautilus*), an internal shell, as in squid (e.g., *Loligo* spp.) and cuttlefish (e.g., *Sepia* spp.), or no shell, as in octopods (e.g., *Octopus*, *Eledone*) [89]. They live pelagically and are almost all fast-swimming carnivores. Only a few of the cephalopod species are commercially harvested on a large scale, with squid being the main species. Some benthic cephalopods might be farmed in marine aquaculture as they show fast growth and have a high market price [89]. 

Among protistan parasites, ciliates are one of the most common cephalopods and coccidia of the genus *Aggregata* (Apicomplexa) is the most widely distributed. Damages caused by these parasites include tissue damage and physiological malfunction, and they may affect the cellular immune response [90]. Normally, coccidiosis is not a fatal disease in cephalopods, but it weakens their innate immunity, making them vulnerable to secondary infections. 

Dicyemida is a phylum of Mesozoan parasites that lives in the renal appendages of cephalopods. Dicyemid (Rhombozoa) infections can be heavy but apparently these infections cause no damage to the host [91]. 

Metazoan parasites of cephalopods including platyhelminths, cestodes, monogeneans digenetic trematodes, nematodes, acanthocephalans, and crustaceans, summarized in Table 2. Cephalopods can serve as second intermediate, paratenic, or final hosts, but never as first intermediate hosts [91]. Metacercariae of didymozoid (Trematoda, Plagiorchiida) are the most important group of digeneans that infects oceanic squids [91]. Larval ascaridoid nematodes are a commonly reported parasitic agent in European cephalopods [91]. Only few studies have been published on crustaceans infecting European cephalopods. Branchiurans and cymothoid isopods have been found on the skin and in the mantle cavity of cuttlefish (*Sepia* and *Sepiola* species) [91].

### 3.4. Crustaceans

Small arthropoda species are cultured as food for young fishes and the main parasites affecting these are various protistan species and fungi microsporidia [92]. Shrimp and other decapod species are primarily raised for human consumption and are an important source of aquatic food protein worldwide [93]. Decapod species are cultured both in marine and freshwater aquaculture [94].

Several eukaryotic parasites are important in crustacean aquaculture (Table 2), but bacteria and viruses are more important [95].

### 3.5. Anuran

With the growing interest in the consumption of white and healthy meats, frog culture has been projected as an alternative source of protein [96,97,98]. Polyculture of frogs and fish also seems to be sustainable [99]. In the US, frog culture may not be profitable. According to the FAO, the total production of *Rana* spp. frogs was 147,800 tonnes live weight [17]. Several parasites including Protista, trematodes, nematodes, and cestodes (Table 2), some of which could be zoonotic, are found in Anurans. In some areas, especially in Asia, *Spirometra* is a neglected food- and water-borne disease but the parasite is found worldwide [100,101,102].

### 3.6. Pisces

Culture of finfishes is the most important aquacultural production (Table 1). Production of inland aquaculture far exceeds that in coastal and marine culture (Table 1). Aquaculture is especially well developed in Asian countries, but is also becoming increasingly important in African and Neotropical countries [17]. Inland aquaculture production far exceeds the amount of inland capture, while the opposite is the case for marine production [17]. A very detailed account of parasites in fishes is given in Noga [23], and recent reviews include that by Buchmann [103]. 

### 3.7. Other

There are several other taxa which are cultured in various areas. For example, there is a production of the Chinese softshell turtle, *Trionyx sinensis* [104,105], and river and lake turtles in Testudinata [106], and they may also harbor parasites (Table 2). In Australia, crocodile farming is a multi-million dollar industry [53] and this is also practiced in several other countries. Parasitic infections reported in crocodiles include Acanthocephala, Hirudinea, nematodes, Digenea, Cestoda, and Pentastomidae [107,108,109,110]. A checklist of host–parasite interactions for crocodilians has been published [111].

According to the FAO [17], there is a substantial production of the Japanese sea cucumber, *Apostichopus japonicas*, and the edible red jellyfish, *Rhopilema esculentum*. This jellyfish is an important fishery source in China [112,113] and has many uses, including pharmaceutical uses [114]. Parasites of sea cucumbers include protozoan, platyhelmiths, and copepods [115,116], whereas we have not found information about parasites of the jellyfish. 

## 4. Zoonotic Parasites

The most common zoonotic parasites include nematodes such as *Angiostrongylus* spp. [117,118], Anisakiidae [119], and trematodes such as Schistosomatidae, Opisthichordiae, Heterophyidae, Fasciolidae, and Paragonimidae [40,120]. Intermediate hosts of some of the other trematodes can thrive in aquacultural ponds or in habitats receiving aquacultural wastewater because they benefit from the nutrient loading [120]. 

Fish-borne zoonotic trematodes are especially important for aquaculture, because the human infective stage lodges in the final product. The load of metacercariae in fishes can be quite high, which impacts production and marketability.

*Schistosoma* spp. may not be important for aquacultural production per se and transmission within aquaculture facilities may be minimal provided public access to these facilities is prevented (e.g., by fencing) and measures are taken to reduce or prevent contamination of the water with parasitic eggs via the provision of sanitary facilities within the aquacultural farms [26]. The schistosome intermediate hosts can thrive in freshwater aquacultural ponds or augment the population density of these hosts in recipient habitats of wastewater due to organic loading. The major problem arises with the wastewater, if it is left untreated into natural habitats where human water contact can be high or if children swim in the aquacultural ponds, as we observed in Africa. The organic loading from the wastewater can greatly contribute to the proliferation of the intermediate snail hosts and thereby increase transmission.

The same principles apply to Fasciolosis, which is caused by *Fasciola hepatica* or *Fasciola gigantica*, in larger domestic animals, e.g., sheep, cattle, and others, as these are not likely roaming around within aquacultural facilities. If untreated manure from infected animals is used for pond fertilization, there is potential for transmission. The intermediate hosts for these trematodes, certain species of the Lymnaeidae, can thrive in aquacultural facilities and thus can become infected, e.g., if feces from final hosts is used to fertilize ponds.

Nematodes, such as *Angiostrongylus* spp., can be found in both land and freshwater snails, wherefore there potentially could be zoonotic problems in areas where snails are consumed that are insufficiently heat treated [121]. Other eukaryotic parasites associated with aquaculture can also be zoonotic [122,123,124].

## 5. Environmental Impact of Aquaculture

The problem of parasitism in aquaculture cannot be seen in isolation. Many factors contribute to aggravating problems with parasitism. Poor water quality can contribute to the proliferation of parasites, especially protistan and fungal species. Stress due to poor quality or feeding status can render aquacultural species more susceptible to infection. Within aquacultural facilities, water can be overloaded with organic material in an attempt to optimize production, and this can lead to poor water conditions that could stress the aquacultured species. In order to improve water conditions and reduce the accumulation of food remnants, multitrophic aquaculture could be practiced, and employing biofloc techniques combined with aquaponics could reduce the need for water replacement and cleaning of the facilities [26]. In multitrophic culture systems, filter feeders such as, for example, gastropods of Viviparidae (freshwater) or bivalves (marine and freshwater) could play an important role in securing good water conditions and thereby affect fishes’ wellbeing [26,125]. Also, some annelid species are important filter feeders [126]. 

Pond fertilization using untreated manure (probably only in freshwater aquaculture) from husbandry could introduce parasitic eggs or pathogenic microorganisms, and this could augment the transmission of parasitic infections in fishes, and in the case of zoonotic parasites or microorganisms, create a public health problem [26].

The accumulation of food remnants in the sediment ponds will have to be removed at intervals, probably at the time of harvesting. This leads to another major problem, i.e., the eutrophication of recipient natural waterbodies. In cage culture, the area below and around cages accumulates remnants and feces; but here, annelid species could contribute to remediation [70,72]. 

In order to minimize environmental impact, circular production systems should be used where waste is seen as a valuable resource rather than waste material [127]. Annelids could be fed sludge from ponds and they in turn could be used as feed for other cultured species, such as fishes and crustaceans [65]. Water from aquacultural ponds, however, may contain metals from feed or medicine residues that might be used in the production; thus, it may be necessary to remove these chemicals from the water, possibly through phyto-remediation [128].

## 6. Aquaculture and the Spread of Parasites or Cultured Species

There are numerous examples where species introduced into an area for aquaculture have become invasive and possibly included some parasitic species [129]. For example, the golden apple snails, *Pomacea canaliculata*, were introduced in Asia for aquaculture and quickly spread to other areas [130]. They are a major pest in rice fields, causing huge damage [131,132,133]; they can destroy native faunas [134], and since they are a host of *Angiostrongylus cantonensis*, they may contribute to aggravating the transmission of this parasite [135,136,137,138]. Recently, *P. canaliculata* was reported in Africa, i.e., from the Mwea Irrigation Scheme [139,140], which is an important area for rice cultivation in Kenya, and this species could have devastating effects on rice production. For freshwater mollusks, the major modes of long-distance dispersal would be the international trade in ornamental plants, fishes, bait trade, or wet-cultivated crops [141,142,143,144]. Also, some species have been introduced deliberately for the biological control of intermediate host snails of schistosomes [145,146]. 

For fish, the main mode of dispersal would be deliberate introduction for aquaculture [147]. The introduction of *Oreochromis niloticus* in Lake Victoria [148], *Oreochromis niloticus* for cage culture in crater lakes in Nicaragua [149], and blue tilapia, *Oreochromis aureas*, in Pennsylvania, US [150], for aquaculture are examples where these fishes have been established in natural habitats. 

## 7. Control of Parasites

The control of parasites in finfish-culture was recently reviewed by Buchmann [103] and includes various chemical or biological treatment options for ectoparasites, using medicinal treatment or vaccination against some parasites. Feed additives may partly control some infections, or selective breeding can produce more disease-resistant fishes [103]. There has been focus on controlling zoonotic parasites associated with aquaculture, especially trematodes and nematodes [136,151]. The control of trematodes, whether zoonotic or not, is often attempted through the control of the first intermediate hosts, snails or bivalves; but other measures are important as well to reduce transmission, such as reducing the inflow of parasitic eggs or reducing the exposure of final hosts (people or reservoir hosts) [151]. For control snails or other intermediate hosts in aquaculture ponds, chemical means may not be an option as these chemicals (e.g., molluscicides) are also toxic to fishes. Hydrated lime, however, has been used to control intermediate hosts of *Bolbophorus damnificus* and *Clinostomum marginatum* along the margins of ponds with *Pangasius* in the US [152]. Therefore, there is need to develop new approaches for controlling essential intermediate hosts of parasites to protect the cultured species from infection. Some of the measures implemented against zoonotic trematodes could also be of value for the control of other parasite-host systems, e.g., preventing parasite eggs from entering the aquaculture facilities or stocking predators or competitors of the intermediate hosts. Since both intermediate hosts and parasites respond to chemical cues in their environment, bait formulations might be developed for the selective removal of these organisms from aquaculture facilities [153,154].

## 8. Conclusions

Clearly, to further develop aquaculture in a sustainable manner one must take a holistic approach where many factors, e.g., human health, food safety, animal health and welfare, environmental and biodiversity protection, and marketability mechanisms, as well as probably other factors, are considered. This approach is often referred to as One Health. As more species are included for aquaculture, new host–parasite relationships may be created, either because of new parasitic species introduced with cultured species or because new aquacultural techniques could be favoring the transmission of parasites. Therefore, much research is needed to find ways to mitigate the problems associated with aquaculture and this should minimize at the same time the environmental impact by implementing circular technologies.

Aquacultural production is strongly dominated by the Asian region, but within each geographical region, production is dominated by a few countries in each geographical area [17], therefore there is still a great possibility of further developing aquacultural production in all continents.

## Figures and Tables

**Figure 1 biology-13-00041-f001:**
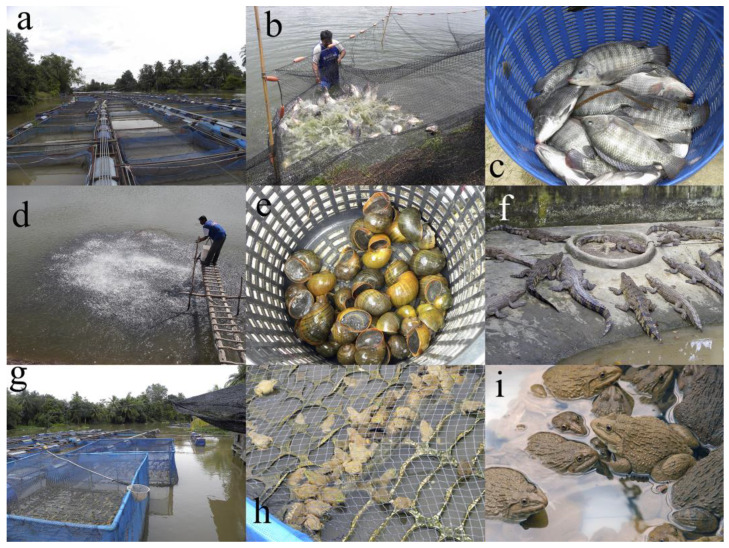
Examples of aquaculture: (**a**) Cage culture of *Oreochromis niloticus*; (**b**) pond culture of *O. niloticus*; (**c**) *O. niloticus*; (**d**) feeding juvenile *Pangasianodon hypophthalmus*; (**e**) harvested apple snails; (**f**) crocodile farm; (**g**,**h**) cage culture of frogs; (**i**) culture of frogs for research.

**Table 2 biology-13-00041-t002:** Overview of combinations of parasite taxa and important aquacultured species. References reviewed for construction of this table can be found in the Appendix A. (+) Only minor effect except for heavy infestation; (++) Important; (+++) Very important.

	Taxa Raised in Aquaculture								
Parasites	Corals	Annelida	Gastropoda	Bivalvia	Cephalopoda	Crustacea	Pisces	Anurans	Turtles
**Protista**									
Apicomplexa					+++	++			
Ascetosporea				+++					
Cercozoa			++	+++	++			++	++
Protozoa			+	++		++	++	++	
Amoebozoa			+					++	
Ciliophora		+	+	+++	++	++	++	++	
Perkinsozoa				+++					
Flagellates		+	+				++		
Dinoflagellata				++		++	++		
**Fungi, etc.**									
Oomycetes						+++	+++	++	
Microsporidia			+	++	++	++	++		
Other			+	+	++	+	++		
**Plantae**									
Green algae			+	+		+	+		++
Other			+						
**Animalia**									
Porifera				++					
Cnidaria				++					
Myxozoa		++					+++	++	
Acoela	++								
Acanthocephala					++	++			
Rhombozoa					+				
Annelida				++			+		
Branchiura					++				
Citellata				++					
Polychaetes				++		++			
Oligochaetes			+	++		++			
Hirudinea			+	++			++		
Nematoda			+	++	+++	++	++	+	
**Platyhelminthes**									
Polycladidia	+								
Turbellaria			+	++		+			
Monogenea					+	+	+++	++	+
Digenea	+	+	+++	+++		+	+++	++	
Cestoda		+		+	+	+	+	++	
Temnocephalidae						+			
Nemertinea				+					
**Mollusca**									
Gastropoda	+	+	+	+					
Bivalvia							+		
**Arthropoda**	+						+	+	
Decapoda				+					
Copepoda, Ostracoda, Isopoda	+	+	+	+	++	++	+++	+	
Branchiura					++		+++		
Cirripedia	+			+					
Acarina Mites				+		+	++		
**Pisces**							+		

**Table 3 biology-13-00041-t003:** Aquaculture production (in 1000 tonnes live weight) in inland and coastal-marine systems by region and major cultured species. Data from the FAO [17].

Cultured Species	Africa	Americas	Asia	Europe	Oceania	World
Inland						
Finfish	1857	1180	45,527	552	5	49,120
Crustaceans	<1	73	4401	3	<1	4477
Mollusks			193			193
Other animals		<1	593	<1		594
Coastal-Marine						
Finfish	379	1241	4503	2122	96	8341
Crustaceans	8	1194	5550	<1	8	6760
Mollusks	6	688	16,159	579	116	17,548
Other animals	<1		459	6	3	469

## Data Availability

Not relevant.

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
