# Peer review of "Aquaculture of Animal Species: Their Eukaryotic Parasites and the Control of Parasitic Infections"

_biology, 2024, doi:10.3390/biology13010041_

Round 1

Reviewer 1 Report

Comments and Suggestions for Authors

My comments to the paper as below:

Overall the paper is interesting and has novelty

However, authors may enhance the manuscript by providing prevention and treatment for parasite infections

as we know parasite need few hosts to complete its life cycle, then, author may elaborate more in this aspect

moreover, any significant impacts such as economical loss, public health and etc due to parasite infection

a good review paper must synthesis new idea from the previous studies findings

author may provide the future work 

Comments on the Quality of English Language

no an issue

Reviewer 2 Report

Comments and Suggestions for Authors

This publication is devoted to a current topic. The development of aquaculture is a new source of impact on ecosystems, the scale of the impact is comparable to the impact of a large industrial enterprise and with the addition of biological threats. On the other hand, places of artificial influence are a consequence - a donor and a source of invasion of both aquaculture objects and the organisms parasitizing on them. If the cultivation and control regime is violated, transmission of pathogens to humans and animals is possible.

Even cultivated European amphibians can cause Alariosis, caused by the trematode Alaria alata. In subtropical and tropical climates, the conditions and number of species of pathogenic organisms are higher, which complicates control and aggravates the consequences of the introduction and invasion of alien organisms.

Thus, the relevance of the study does not trigger the proposal. This will require the following changes:

1. Setting the goal and objectives of the study

2. The authors have carried out a large analysis of the available data, but their generalized presentation is necessary. For example, present a register of types and main aquaculture objects, with an analysis of the degree of risk for terrestrial and coastal marine ecosystems (Table 1), taking into account distribution by continent (Table 3), by factors - “human health, food safety, animal health and welfare, environmental and biodiversity protection".

3. Submission of data in Tabl format. 3 raises big questions: “Table 2. Selected references for various combinations of parasite taxa and important aquacultured 116 species.” I believe that it is necessary to indicate the number or range of species (taxa).

In general, the material presented in the article “Aquaculture of animal species: Their eukaryotic parasites and 2 control of parasitic infections” requires a clearer and more understandable structure. It is necessary to finalize the design, a number of tables and generalizations to be presented as diagrams, for example, information from “Table 3. Aquaculture production (in 1000 tons live weight) in inland and coastal-marine systems  by region and major cultured species. Data from FAO [17].

Round 2

Reviewer 2 Report

Comments and Suggestions for Authors

The authors generally agree with the comments, but not all were reflected in the text of the manuscript.

Table line 129 (“Table 2. Selected references for various combinations of parasite taxa and important aquaculture species.”), indicating not data from sources, such as the number of taxa in a sequential or other symbol such as “+”, but direct references to the Source does not reveal information, but is essentially literary or background information.

This information reflects the degree to which the problem has been studied and can be placed in additional materials.

Some comments.

1. Table 2. Selected references for various combinations of parasite taxa and important aquacultured species" requires a design change.

2. Line 735, requires modification “van den Berg, A.H., et al., The impact of the water molds Saprolegnia diclina and Saprolegnia parasitica on natural ecosystems and the aquaculture industry. Fungal Biology Reviews, 2013. 27(2): p. 33-42."

I believe that the article should pay attention to the most pathogenic and potential parasites in terms of the number of taxa of aquaculture objects by continent, taking into account climatic conditions, which will reveal the value of “Table 3. Aquaculture production (in 1000 tons live weight) in inland and coastal-marine systems by region and major cultured species. Data from FAO [17]. This addition, although it will take some time from the authors, will make the presented work a logically completed study.

The article has undergone changes, the text has been improved, the reviewer thanks for the corrections, but some revision is required.
